# RIEMANNIAN OPTIMIZATION FOR SKIP-GRAM NEGATIVE SAMPLING

**Alexander Fonarev**[123]**, Alexey Grinchuk**[12]**, Gleb Gusev**[2]**, Pavel Serdyukov**[2]**, Ivan Oseledets**[14]
[1]Skolkovo Institute of Science and Technology, Moscow, Russia
[2]Yandex LLC, Moscow, Russia
[3]SBDA Group, Dublin, Ireland
[4]Institute of Numerical Mathematics, Russian Academy of Sciences, Moscow, Russia
`newo@newo.su, oleksii.hrinchuk@skolkovotech.ru, gleb57@yandex-team.ru,`
`pavser@yandex-team.ru, ioseledets@skoltech.ru`

## ABSTRACT

Skip-Gram Negative Sampling (SGNS) word embedding model, well known by its implementation in "word2vec" software, is usually optimized by stochastic gradient descent. It can be shown that optimizing for SGNS objective can be viewed as an optimization problem of searching for a good matrix with the low-rank constraint. The most standard way to solve this type of problems is to apply Riemannian optimization framework to optimize the SGNS objective over the manifold of required low-rank matrices. In this paper, we propose an algorithm that optimizes SGNS objective using Riemannian optimization and demonstrates its superiority over popular competitors, such as the original method to train SGNS and SVD over SPPMI matrix.

## 1 INTRODUCTION

In this paper, we consider the problem of embedding words into a low-dimensional space in order to measure the semantic similarity between them. As an example, how to find whether the word "table" is semantically more similar to the word "stool" than to the word "sky"? That is achieved by constructing a low-dimensional vector representation for each word and measuring similarity between the words as the similarity between the corresponding vectors.

One of the most popular word embedding models by Mikolov et al. (2013) is a discriminative neural network that optimizes Skip-Gram Negative Sampling (SGNS) objective (see Equation 3). It aims at predicting whether two words can be found close to each other within a text. As shown in Section 2, the process of word embeddings training using SGNS can be divided into two general steps with clear objectives:

**Step 1.** Search for a low-rank matrix $X$ that provides a good SGNS objective value;

**Step 2.** Search for a good low-rank representation $X = WC^\top$ in terms of linguistic metrics, where $W$ is a matrix of word embeddings and $C$ is a matrix of so-called context embeddings.

Unfortunately, most previous approaches mixed these two steps into a single one, what entails a not completely correct formulation of the optimization problem. For example, popular approaches to train embeddings (including the original "word2vec" implementation) do not take into account that the objective from Step 1 depends only on the product $X = WC^\top$: instead of straightforward computing of the derivative w.r.t. $X$, these methods are explicitly based on the derivatives w.r.t. $W$ and $C$, what complicates the optimization procedure. Moreover, such approaches do not take into account that parametrization $WC^\top$ of matrix $X$ is non-unique and Step 2 is required. Indeed, for any invertible matrix $S$, we have $X = W_1 C_1^\top = W_1 S S^{-1} C_1^\top = W_2 C_2^\top$, therefore, solutions $W_1 C_1$ and $W_2 C_2$ are equally good in terms of the SGNS objective but entail different cosine similarities between embeddings and, as a result, different performance in terms of linguistic metrics (see Section 4.2 for details).

A successful attempt to follow the above described steps, which outperforms the original SGNS optimization approach in terms of various linguistic tasks, was proposed by Levy & Goldberg (2014). In order to obtain a low-rank matrix $X$ on Step 1, the method reduces the dimensionality of Shifted Positive Pointwise Mutual Information (SPPMI) matrix via Singular Value Decomposition (SVD). On Step 2, it computes embeddings $W$ and $C$ via a simple formula that depends on the factors obtained by SVD. However, this method has one important limitation: SVD provides a solution to a surrogate optimization problem, which has no direct relation to the SGNS objective. In fact, SVD minimizes the Mean Squared Error (MSE) between $X$ and SPPMI matrix, what does not lead to minimization of SGNS objective in general (see Section 6.1 and Section 4.2 in Levy & Goldberg (2014) for details).

These issues bring us to the main idea of our paper: while keeping the low-rank matrix search setup on Step 1, optimize the original SGNS objective directly. This leads to an optimization problem over matrix $X$ with the low-rank constraint, which is often (Mishra et al. (2014)) solved by applying *Riemannian optimization* framework (Udriste (1994)). In our paper, we use the projector-splitting algorithm (Lubich & Oseledets (2014)), which is easy to implement and has low computational complexity. Of course, Step 2 may be improved as well, but we regard this as a direction of future work.

As a result, our approach achieves the significant improvement in terms of SGNS optimization on Step 1 and, moreover, the improvement on Step 1 entails the improvement on Step 2 in terms of linguistic metrics. That is why, the proposed two-step decomposition of the problem makes sense, what, most importantly, opens the way to applying even more advanced approaches based on it (e.g., more advanced Riemannian optimization techniques for Step 1 or a more sophisticated treatment of Step 2).

To summarize, the main contributions of our paper are:

- We reformulated the problem of SGNS word embedding learning as a two-step procedure with clear objectives;
- For Step 1, we developed an algorithm based on Riemannian optimization framework that optimizes SGNS objective over low-rank matrix $X$ directly;
- Our algorithm outperforms state-of-the-art competitors in terms of SGNS objective and the semantic similarity linguistic metric (Levy & Goldberg (2014); Mikolov et al. (2013); Schnabel et al. (2015)).

## 2 PROBLEM SETTING

### 2.1 SKIP-GRAM NEGATIVE SAMPLING

In this paper, we consider the Skip-Gram Negative Sampling (SGNS) word embedding model (Mikolov et al. (2013)), which is a probabilistic discriminative model. Assume we have a text corpus given as a sequence of words $w_1, \ldots, w_n$, where $n$ may be larger than $10^{12}$ and $w_i \in V_W$ belongs to a vocabulary of words $V_W$. A *context* $c \in V_C$ of the word $w_i$ is a word from set $\{w_{i-L}, ..., w_{i-1}, w_{i+1}, ..., w_{i+L}\}$ for some fixed window size $L$. Let $\mathbf{w}, \mathbf{c} \in \mathbb{R}^d$ be the *word embeddings* of word $w$ and context $c$, respectively. Assume they are specified by the following mappings:

$$\mathcal{W} : V_W \to \mathbb{R}^d, \quad \mathcal{C} : V_C \to \mathbb{R}^d.$$

The ultimate goal of SGNS word embedding training is to fit good mappings $\mathcal{W}$ and $\mathcal{C}$.

In the SGNS model, the probability that pair $(w, c)$ is observed in the corpus is modeled as a following function:

$$P\left((w, c) \in D | w, c\right) = \sigma(\langle \mathbf{w}, \mathbf{c} \rangle) = \frac{1}{1 + \exp(-\langle \mathbf{w}, \mathbf{c} \rangle)}, \tag{1}$$

where $D$ is the multiset of all word-context pairs $(w, c)$ observed in the corpus and $\langle \mathbf{x}, \mathbf{y} \rangle$ is the scalar product of vectors $\mathbf{x}$ and $\mathbf{y}$. Number $d$ is a hyperparameter that adjusts the flexibility of the model. It usually takes values from tens to hundreds.

In order to collect a training set, we take all pairs $(w, c)$ from $D$ as positive examples and $k$ randomly generated pairs $(w, c)$ as negative ones. Let $\#(w, c)$ be the number of times the pair $(w, c)$ appears

in $D$. Thereby the number of times the word $w$ and the context $c$ appear in $D$ can be computed as $\#(w) = \sum_{c \in V_c} \#(w, c)$ and $\#(c) = \sum_{w \in V_w} \#(w, c)$ accordingly. Then negative examples are generated from the distribution defined by $\#(c)$ counters: $P_D(c) = \frac{\#(c)}{|D|}$. In this way, we have a model maximizing the following logarithmic likelihood objective for each word pair $(w, c)$:

$$\#(w, c)(\log \sigma(\langle \mathbf{w}, \mathbf{c} \rangle) + k \cdot \mathbb{E}_{c' \sim P_D} \log \sigma(-\langle \mathbf{w}, \mathbf{c}' \rangle)). \tag{2}$$

In order to maximize the objective over all observations for each pair $(w, c)$, we arrive at the following SGNS optimization problem over all possible mappings $\mathcal{W}$ and $\mathcal{C}$:

$$l = \sum_{w \in V_W} \sum_{c \in V_C} \#(w, c)(\log \sigma(\langle \mathbf{w}, \mathbf{c} \rangle) + k \cdot \mathbb{E}_{c' \sim P_D} \log \sigma(-\langle \mathbf{w}, \mathbf{c}' \rangle)) \to \max_{\mathcal{W},} . \tag{3}$$

Usually, this optimization is done via the stochastic gradient descent procedure that is performed during passing through the corpus (Mikolov et al. (2013); Rong (2014)).

## 2.2 OPTIMIZATION OVER LOW-RANK MATRICES

Relying on the prospect proposed by Levy & Goldberg (2014), let us show that the optimization problem given by (3) can be considered as a problem of searching for a matrix that maximizes a certain objective function and has the rank-$d$ constraint (Step 1 in the scheme described in Section 1).

### 2.2.1 SGNS LOSS FUNCTION

As shown by Levy & Goldberg (2014), the logarithmic likelihood (3) can be represented as the sum of $l_{w,c}(\mathbf{w}, \mathbf{c})$ over all pairs $(w, c)$, where $l_{w,c}(\mathbf{w}, \mathbf{c})$ has the following form:

$$l_{w,c}(\mathbf{w}, \mathbf{c}) = \#(w, c) \log \sigma(\langle \mathbf{w}, \mathbf{c} \rangle) + k \frac{\#(w)\#(c)}{|D|} \log \sigma(-\langle \mathbf{w}, \mathbf{c} \rangle). \tag{4}$$

A crucial observation is that this loss function depends only on the scalar product $\langle \mathbf{w}, \mathbf{c} \rangle$ but not on embeddings $\mathbf{w}$ and $\mathbf{c}$ separately:

$$l_{w,c}(\mathbf{w}, \mathbf{c}) = f_{w,c}(x_{w,c}),$$
$$f_{w,c}(x_{w,c}) = a_{w,c} \log \sigma(x_{w,c}) + b_{w,c} \log \sigma(-x_{w,c}),$$

where $x_{w,c}$ is the scalar product $\langle \mathbf{w}, \mathbf{c} \rangle$ and $a_{w,c} = \#(w, c)$, $b_{w,c} = k \frac{\#(w)\#(c)}{|D|}$ are constants.

### 2.2.2 MATRIX NOTATION

Denote $|V_W|$ as $n$ and $|V_C|$ as $m$. Let $W \in \mathbb{R}^{n \times d}$ and $C \in \mathbb{R}^{m \times d}$ be matrices, where each row $\mathbf{w} \in \mathbb{R}^d$ of matrix $W$ is the word embedding of the corresponding word $w$ and each row $\mathbf{c} \in \mathbb{R}^d$ of matrix $C$ is the context embedding of the corresponding context $c$. Then the elements of the product of these matrices

$$X = WC^\top$$

are the scalar products $x_{w,c}$ of all pairs $(w, c)$:

$$X = (x_{w,c}), \quad w \in V_W, c \in V_C.$$

Note that this matrix has rank $d$, because $X$ equals to the product of two matrices with sizes $(n \times d)$ and $(d \times m)$. Now we can write SGNS objective given by (3) as a function of $X$:

$$F(X) = \sum_{w \in V_W} \sum_{c \in V_C} f_{w,c}(x_{w,c}), \quad F: \mathbb{R}^{n \times m} \to \mathbb{R}. \tag{5}$$

This arrives us at the following proposition:

**Proposition 1** *SGNS optimization problem given by* (3) *can be rewritten in the following constrained form:*

$$\begin{aligned} \underset{X \in \mathbb{R}^{n \times m}}{maximize} \quad & F(X), \\ subject\ to \quad & X \in \mathcal{M}_d, \end{aligned} \tag{6}$$

*where $\mathcal{M}_d$ is the manifold (Udriste (1994)) of all matrices in $\mathbb{R}^{n \times m}$ with rank $d$:*

$$\mathcal{M}_d = \{X \in \mathbb{R}^{n \times m} : rank(X) = d\}.$$

The key idea of this paper is to solve the optimization problem given by (6) via the framework of Riemannian optimization, which we introduce in Section 3.

Important to note that this prospect does not suppose the optimization over parameters $W$ and $C$ directly. This entails the optimization in the space with $((n + m - d) \cdot d)$ degrees of freedom (Mukherjee et al. (2015)) instead of $((n + m) \cdot d)$, what simplifies the optimization process (see Section 5 for the experimental results).

## 2.3   COMPUTING EMBEDDINGS FROM A LOW-RANK SOLUTION

Once $X$ is found, we need to recover $W$ and $C$ such that $X = WC^\top$ (Step 2 in the scheme described in Section 1). This problem does not have a unique solution, since if $(W, C)$ satisfy this equation, then $WS^{-1}$ and $CS^\top$ satisfy it as well for any non-singular matrix $S$. Moreover, different solutions may achieve different values of the linguistic metrics (see Section 4.2 for details). While our paper focuses on Step 1, we use, for Step 2, a heuristic approach that was proposed by Levy et al. (2015) and it shows good results in practice. We compute SVD of $X$ in the form $X = U\Sigma V^\top$, where $U$ and $V$ have orthonormal columns, and $\Sigma$ is the diagonal matrix, and use

$$W = U\sqrt{\Sigma}, \quad C = V\sqrt{\Sigma}$$

as matrices of embeddings.

A simple justification of this solution is the following: we need to map words into vectors in a way that similar words would have similar embeddings in terms of cosine similarities:

$$\cos(\mathbf{w}_1, \mathbf{w}_2) = \frac{\langle \mathbf{w}_1, \mathbf{w}_2 \rangle}{\|\mathbf{w}_1\| \cdot \|\mathbf{w}_2\|}.$$

It is reasonable to assume that two words are similar, if they share contexts. Therefore, we can estimate the similarity of two words $w_1$, $w_2$ as $s(w_1, w_2) = \sum_{c \in V_C} x_{w_1,c} \cdot x_{w_2,c}$, what is the element of the matrix $XX^\top$ with indices $(w_1, w_2)$. Note that $XX^\top = U\Sigma V^\top V\Sigma U^\top = U\Sigma^2 U^\top$. If we choose $W = U\Sigma$, we exactly obtain $\langle \mathbf{w_1}, \mathbf{w_2} \rangle = s(w_1, w_2)$, since $WW^\top = XX^\top$ in this case. That is, the cosine similarity of the embeddings $\mathbf{w_1}$, $\mathbf{w_2}$ coincides with the intuitive similarity $s(w_1, w_2)$. However, scaling by $\sqrt{\Sigma}$ instead of $\Sigma$ was shown by Levy et al. (2015) to be a better solution in experiments.

## 3   PROPOSED METHOD

### 3.1   RIEMANNIAN OPTIMIZATION

#### 3.1.1   GENERAL SCHEME

The main idea of Riemannian optimization (Udriste (1994)) is to consider (6) as a constrained optimization problem. Assume we have an approximated solution $X_i$ on a current step of the optimization process, where $i$ is the step number. In order to improve $X_i$, the next step of the standard gradient ascent outputs $X_i + \nabla F(X_i)$, where $\nabla F(X_i)$ is the gradient of objective $F$ at the point $X_i$. Note that the gradient $\nabla F(X_i)$ can be naturally considered as a matrix in $\mathbb{R}^{n \times m}$. Point $X_i + \nabla F(X_i)$ leaves the manifold $\mathcal{M}_d$, because its rank is generally greater than $d$. That is why Riemannian optimization methods map point $X_i + \nabla F(X_i)$ back to manifold $\mathcal{M}_d$. The standard Riemannian gradient method first projects the gradient step onto the tangent space at the current point $X_i$ and then *retracts* it back to the manifold:

$$X_{i+1} = R\left(P_{\mathcal{T}_M}(X_i + \nabla F(X_i))\right),$$

where $R$ is the *retraction* operator, and $P_{\mathcal{T}_M}$ is the projection onto the tangent space.

#### 3.1.2   PROJECTOR-SPLITTING ALGORITHM

In our paper, we use a much simpler version of such approach that retracts point $X_i + \nabla F(X_i)$ directly to the manifold, as illustrated on Figure 1: $X_{i+1} = R(X_i + \nabla F(X_i))$.

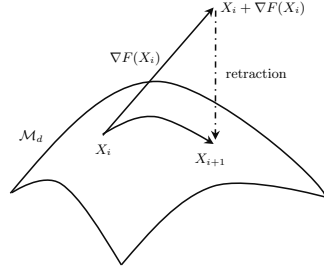

Figure 1: Geometric interpretation of one step of projector-splitting optimization procedure: the gradient step an the retraction of the high-rank matrix $X_i + \nabla F(X_i)$ to the manifold of low-rank matrices $\mathcal{M}_d$.

Intuitively, retractor $R$ finds a rank-$d$ matrix on the manifold $\mathcal{M}_d$ that is similar to high-rank matrix $X_i + \nabla F(X_i)$ in terms of Frobenius norm. How can we do it? The most straightforward way to reduce the rank of $X_i + \nabla F(X_i)$ is to perform the SVD, which keeps $d$ largest singular values of it:

$$
\begin{aligned}
&1: U_{i+1}, S_{i+1}, V_{i+1}^\top \leftarrow \text{SVD}(X_i + \nabla F(X_i)), \\
&2: X_{i+1} \leftarrow U_{i+1} S_{i+1} V_{i+1}^\top.
\end{aligned}
\tag{7}
$$

However, it is computationally expensive. Instead of this approach, we use the projector-splitting method (Lubich & Oseledets (2014)), which is a second-order retraction onto the manifold (for details, see the review by Absil & Oseledets (2015)). Its practical implementation is also quite intuitive: instead of computing the full SVD of $X_i + \nabla F(X_i)$ according to the gradient projection method, we use just one step of the block power numerical method (Bentbib & Kanber (2015)) which computes the SVD, what reduces the computational complexity.

Let us keep the current point in the following factorized form:

$$
X_i = U_i S_i V_i^\top,
\tag{8}
$$

where matrices $U_i \in \mathbb{R}^{n \times d}$ and $V_i \in \mathbb{R}^{m \times d}$ have $d$ orthonormal columns and $S_i \in \mathbb{R}^{d \times d}$. Then we need to perform two QR-decompositions to retract point $X_i + \nabla F(X_i)$ back to the manifold:

$$
\begin{aligned}
&1: U_{i+1}, S_{i+1} \leftarrow \text{QR}\left((X_i + \nabla F(X_i))V_i\right), \\
&2: V_{i+1}, S_{i+1}^\top \leftarrow \text{QR}\left((X_i + \nabla F(X_i))^\top U_{i+1}\right), \\
&3: X_{i+1} \leftarrow U_{i+1} S_{i+1} V_{i+1}^\top.
\end{aligned}
$$

In this way, we always keep the solution $X_{i+1} = U_{i+1} S_{i+1} V_{i+1}^\top$ on the manifold $\mathcal{M}_d$ and in the form (8).

What is important, we only need to compute $\nabla F(X_i)$, so the gradients with respect to $U$, $S$ and $V$ are never computed explicitly, thus avoiding the subtle case where $S$ is close to singular (so-called singular (critical) point on the manifold). Indeed, the gradient with respect to $U$ (while keeping the orthogonality constraints) can be written (Koch & Lubich (2007)) as:

$$
\frac{\partial F}{\partial U} = \frac{\partial F}{\partial X} V S^{-1},
$$

which means that the gradient will be large if $S$ is close to singular. The projector-splitting scheme is free from this problem.

## 3.2 ALGORITHM

In case of SGNS objective given by (5), an element of gradient $\nabla F$ has the form:

$$
(\nabla F(X))_{w,c} = \frac{\partial f_{w,c}(x_{w,c})}{\partial x_{w,c}} = \#(w, c) \cdot \sigma\left(-x_{w,c}\right) - k \frac{\#(w)\#(c)}{|D|} \cdot \sigma\left(x_{w,c}\right).
$$

To make the method more flexible in terms of convergence properties, we additionally use $\lambda \in \mathbb{R}$, which is a step size parameter. In this case, retractor $R$ returns $X_i + \lambda \nabla F(X_i)$ instead of $X_i + \nabla F(X_i)$ onto the manifold.

The whole optimization procedure is summarized in Algorithm 1.

---

**Algorithm 1** Riemannian Optimization for SGNS

---

**Require:** Dimentionality $d$, initialization $W_0$ and $C_0$, step size $\lambda$, gradient function $\nabla F : \mathbb{R}^{n \times m} \to \mathbb{R}^{n \times m}$, number of iterations $K$
**Ensure:** Factor $W \in \mathbb{R}^{n \times d}$
 1: $X_0 \leftarrow W_0 C_0^\top$                                                       # get an initial point at the manifold
 2: $U_0, S_0, V_0^\top \leftarrow \text{SVD}(X_0)$            # compute the first point satisfying the low-rank constraint
 3: $i \leftarrow 0$
 4: **while** $i < K$ **do**
 5:     $U_{i+1}, S_{i+1} \leftarrow \text{QR}\left((X_i + \lambda \nabla F(X_i))V_i\right)$        # perform one step of the block power method
        with two QR-decompositions
 6:     $V_{i+1}, S_{i+1}^\top \leftarrow \text{QR}\left((X_i + \lambda \nabla F(X_i))^\top U_{i+1}\right)$
 7:     $X_{i+1} \leftarrow U_{i+1} S_{i+1} V_{i+1}^\top$                                    # update the point at the manifold
 8:     $i \leftarrow i + 1$
 9: **end while**
10: $U, \Sigma, V^\top \leftarrow \text{SVD}(X_K)$
11: $W \leftarrow U\sqrt{\Sigma}$                                                      # compute word embeddings
12: **return** $W$

---

## 4 EXPERIMENTAL SETUP

### 4.1 TRAINING MODELS

We compare our method ("RO-SGNS" in the tables) performance to two baselines: SGNS embeddings optimized via Stochastic Gradient Descent, implemented in the original "word2vec", ("SGD-SGNS" in the tables) by Mikolov et al. (2013) and embeddings obtained by SVD over SPPMI matrix ("SVD-SPPMI" in the tables) by Levy & Goldberg (2014). We have also experimented with the blockwise alternating optimization over factors W and C, but the results are almost the same to SGD results, that is why we do not to include them into the paper. The source code of our experiments is available online[1].

The models were trained on English Wikipedia "enwik9" corpus[2], which was previously used in most papers on this topic. Like in previous studies, we counted only the words which occur more than 200 times in the training corpus (Levy & Goldberg (2014); Mikolov et al. (2013)). As a result, we obtained a vocabulary of 24292 unique tokens (set of words $V_W$ and set of contexts $V_C$ are equal). The size of the context window was set to 5 for all experiments, as it was done by Levy & Goldberg (2014); Mikolov et al. (2013). We conduct two series of experiments: for dimensionality $d = 100$ and $d = 200$.

Optimization step size is chosen to be small enough to avoid huge gradient values. However, thorough choice of $\lambda$ does not result in a significant difference in performance (this parameter was tuned on the training data only, the exact values used in experiments are reported below).

### 4.2 EVALUATION

We evaluate word embeddings via the word similarity task. We use the following popular datasets for this purpose: "wordsim-353" (Finkelstein et al. (2001); 3 datasets), "simlex-999" (Hill et al. (2016)) and "men" (Bruni et al. (2014)). Original "wordsim-353" dataset is a mixture of the word pairs for both word similarity and word relatedness tasks. This dataset was split (Agirre et al. (2009)) into two intersecting parts: "wordsim-sim" ("ws-sim" in the tables) and "wordsim-rel" ("ws-rel" in the tables) to separate the words from different tasks. In our experiments, we use both of them on a par with the full version of "wordsim-353" ("ws-full" in the tables). Each dataset contains word pairs together with assessor-assigned similarity scores for each pair. As a quality measure, we use Spearman's correlation between these human ratings and cosine similarities for each pair. We call this quality metric *linguistic* in our paper.

---

[1]https://github.com/newozz/riemannian_sgns
[2]Enwik9 corpus can be found here: http://mattmahoney.net/dc/textdata

|          | $d = 100$            | $d = 200$            |
|----------|----------------------|----------------------|
| SGD-SGNS | $-1.68 \cdot 10^9$   | $-1.67 \cdot 10^9$   |
| SVD-SPPMI| $-1.65 \cdot 10^9$   | $-1.65 \cdot 10^9$   |
| RO-SGNS  | $\mathbf{-1.44 \cdot 10^9}$ | $\mathbf{-1.43 \cdot 10^9}$ |

Table 1: Comparison of SGNS values obtained by the models. The larger is better.

| Dim. $d$  | Algorithm | ws-sim | ws-rel | ws-full | simlex | men |
|-----------|-----------|--------|--------|---------|--------|-----|
|           | SGD-SGNS  | 0.719  | 0.570  | 0.662   | 0.288  | 0.645 |
| $d = 100$ | SVD-SPPMI | 0.722  | 0.585  | 0.669   | 0.317  | **0.686** |
|           | RO-SGNS   | **0.729** | **0.597** | **0.677** | **0.322** | 0.683 |
|           | SGD-SGNS  | 0.733  | 0.584  | 0.677   | 0.317  | 0.664 |
| $d = 200$ | SVD-SPPMI | 0.747  | 0.625  | 0.694   | 0.347  | **0.710** |
|           | RO-SGNS   | **0.757** | **0.647** | **0.709** | **0.353** | 0.701 |

Table 2: Comparison of the methods in terms of the semantic similarity task. Each entry represents the Spearman's correlation between predicted similarities and the manually assessed ones.

## 5 Results of Experiments

First of all, we compare the value of SGNS objective obtained by the methods. The comparison is demonstrated in Table 1.

We see that SGD-SGNS and SVD-SPPMI methods provide quite similar results, however, the proposed method obtains significantly better SGNS values, what proves the feasibility of using Riemannian optimization framework in SGNS optimization problem. It is interesting to note that SVD-SPPMI method, which does not optimize SGNS objective directly, obtains better results than SGD-SGNS method, which aims at optimizing SGNS. This fact additionally confirms the idea described in Section 2.2.2 that the independent optimization over parameters $W$ and $C$ may decrease the performance.

However, the target performance measure of embedding models is the correlation between semantic similarity and human assessment (Section 4.2). Table 2 presents the comparison of the methods in terms of it. We see that our method outperforms the competitors on all datasets except for "men" dataset where it obtains slightly worse results. Moreover, it is important that the higher dimension entails higher performance gain of our method in comparison to the competitors.

In order to understand how exactly our model improves or degrades the performance in comparison to the baseline, we found several words, whose neighbors in terms of cosine distance change significantly. Table 3 demonstrates neighbors of words "five", "he" and "main" in terms of our model and its nearest competitor according to the similarity task — SVD-SPPMI. These words were chosen as representative examples whose neighborhoods in terms of SVD-SPPMI and RO-SGNS models are strikingly different. A neighbour of a source word is bold if we suppose that it has a similar semantic meaning to the source word. First of all, we notice that our model produces much better neighbors of the words describing digits or numbers (see word "five" as an example). The similar situation happens for many other words, e.g. in case of word "main" — the nearest neighbors contain 4 similar words in case of our model instead of 2 in case of SVD-SPPMI. The neighbourhood of word "he" contains less semantically similar words in case of our model. However, it filters out completely irrelevant words, such as "promptly" and "dumbledore".

Talking about the optimal number $K$ of iterations in the optimization procedure and step size $\lambda$, we found that they depend on the particular value of dimensionality $d$. For $d = 100$, we have $K = 25, \lambda \approx 5 \cdot 10^{-5}$, and for $d = 200$, we have $K = 13, \lambda = 10^{-4}$. Moreover, it is interesting that the best results were obtained when SVD-SPPMI embeddings were used as an initialization of Riemannian optimization process.

## 6 Related Work

### 6.1 Word Embeddings

Skip-Gram Negative Sampling was introduced by Mikolov et al. (2013). The "negative sampling" approach was thoroughly is described by Goldberg & Levy (2014), and the learning method is ex-

| five | | | | he | | | | main | | | |
|---|---|---|---|---|---|---|---|---|---|---|---|
| SVD-SPPMI | | RO-SGNS | | SVD-SPPMI | | RO-SGNS | | SVD-SPPMI | | RO-SGNS | |
| Neighbors | Dist. | Neighbors | Dist. | Neighbors | Dist. | Neighbors | Dist. | Neighbors | Dist. | Neighbors | Dist. |
| lb | 0.748 | **four** | 0.999 | **she** | 0.918 | when | 0.904 | **major** | 0.631 | **major** | 0.689 |
| kg | 0.731 | **three** | 0.999 | was | 0.797 | had | 0.903 | busiest | 0.621 | **important** | 0.661 |
| mm | 0.670 | **six** | 0.997 | promptly | 0.742 | was | 0.901 | **principal** | 0.607 | line | 0.631 |
| mk | 0.651 | **seven** | 0.997 | having | 0.731 | who | 0.892 | nearest | 0.607 | external | 0.624 |
| lbf | 0.650 | **eight** | 0.996 | dumbledore | 0.731 | **she** | 0.884 | connecting | 0.591 | **principal** | 0.618 |
| per | 0.644 | and | 0.985 | **him** | 0.730 | by | 0.880 | linking | 0.588 | **primary** | 0.612 |

Table 3: Examples of the semantic neighbors obtained for words "five", "he" and "main" by our method and SVD-SPPMI.

plained by Rong (2014). There are several open-source implementations of SGNS neural network, which is widely known as "word2vec" [34].

As shown in Section 2.2, Skip-Gram Negative Sampling optimization can be reformulated as a problem of searching for a low-rank matrix. In order to be able to use out-of-the-box SVD for this task, Levy & Goldberg (2014) used the surrogate version of SGNS as the objective function. There are two general assumptions made in their algorithm that distinguish it from the SGNS optimization:

1. SVD optimizes Mean Squared Error (MSE) objective instead of SGNS loss function.

2. In order to avoid infinite elements in SPMI matrix, it is transformed in ad-hoc manner (SPPMI matrix) before applying SVD.

This makes the objective not interpretable in terms of the original task (3). As mentioned by Levy & Goldberg (2014), SGNS objective weighs different $(w, c)$ pairs differently, unlike the SVD, which works with the same weight for all pairs, what may entail the performance fall. The comprehensive explanation of the relation between SGNS, SPPMI, SVD-over-SPPMI methods is provided by Keerthi et al. (2015). Lai et al. (2015); Levy et al. (2015) give a good overview of highly practical methods to improve these word embedding models.

## 6.2 RIEMANNIAN OPTIMIZATION

An introduction to optimization over Riemannian manifolds can be found in the paper of Udriste (1994). The overview of retractions of high rank matrices to low-rank manifolds is provided by Absil & Oseledets (2015). The projector-splitting algorithm was introduced by Lubich & Oseledets (2014), and also was mentioned by Absil & Oseledets (2015) as "Lie-Trotter retraction".

Riemannian optimization is succesfully applied to various data science problems: for example, matrix completion (Vandereycken (2013)), large-scale recommender systems (Tan et al. (2014)), and tensor completion (Kressner et al. (2014)).

## 7 CONCLUSIONS AND FUTURE WORK

In our paper, we proposed the general two-step scheme of training SGNS word embedding model and introduced the algorithm that performs the search of a solution in the low-rank form via Riemannian optimization framework. We also demonstrated the superiority of the proposed method, by providing the experimental comparison to the existing state-of-the-art approaches.

It seems to be an interesting direction of future work to apply more advanced optimization techniques to Step 1 of the scheme proposed in Section 1 and to explore the Step 2 — obtaining embeddings with a given low-rank matrix.

---

[3]Original Google word2vec: https://code.google.com/archive/p/word2vec/
[4]Gensim word2vec: https://radimrehurek.com/gensim/models/word2vec.html

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
