# Peer review of "Riemannian Optimization for Skip-Gram Negative Sampling"

_ICLR 2017 — rejected_

[Official Review · AnonReviewer2 · rating 6 · confidence 3 · 14 Dec 2016]
**Elegant method, not sure about the practical benefits**

This paper presents a principled optimization method for SGNS (word2vec).

While the proposed method is elegant from a theoretical perspective, I am not sure what the tangible benefits of this approach are. For example, does using Riemannian optimization allow the model to converge faster than the alternatives? The evaluation doesn't show a dramatic advantage to RO-SGNS; the 1% difference on the word similarity benchmarks is within the range of hyperparameter effects (see "Improving Distributional Similarity with Lessons Learned from Word Embeddings", (Levy et al., 2015)). The theoretical connection to Riemannian optimization is nice though, and it might be useful for understanding related methods in the future.

[Official Review · AnonReviewer3 · rating 5 · confidence 3 · 16 Dec 2016]
**still somewhat confused**

Dear authors,

The authors' response clarified some of my confusion. But I still have the following question:

-- The response said a first contribution is a different formulation: you divide the word embedding learning into two steps, step 1 looks for a low-rank X (by Riemannian optimization), step 2 factorizes X into two matrices (W, C). You are claiming that your model outperforms previous approaches that directly optimizes over (W, C). But since the end result (the factors) is the same, can the authors provide some intuition and justification why the proposed method works better?

As far as I can see, though parameterized differently, the first step of your method and previous methods (SGD) are both optimizing over low-rank matrices. Admittedly, Riemannian optimization avoids the rotational degree of freedom (the invertible matrix S you are mentioning in sec 2.3), but I am not 100% certain at this point this is the source of your gain; learning curves of objectives would help to see if Riemannian optimization is indeed more effective. 

-- Another detail I could not easily find is the following. You said a disadvantage of other approaches is that their factors W and C do not directly reflect similarity. Did you try to multiply the factors W and C from other optimizers and then factorize the product using the method in section 2.3, and use the new W for your downstream tasks? I am not sure if this would cause much difference in the performance.

Overall, I think it is always interesting to apply advanced optimization techniques to machine learning problems. The current paper would be stronger from the machine learning perspective, if more thorough comparison and discussion (as mentioned above) are provided. On the other hand, my expertise is not in NLP and I leave it to other reviewers to decide the significance in experimental results.

[Official Review · AnonReviewer1 · rating 4 · confidence 4 · 28 Dec 2016]
**Not convincing**

The paper considers Grassmannian SGD to optimize the skip gram negative sampling (SGNS) objective for learning better word embeddings. It is not clear why the proposed optimization approach has any advantage over the existing vanilla SGD-based approach - neither approach comes with theoretical guarantees - the empirical comparisons show marginal improvements. Furthermore, the key idea here - that of projector splitting algorithm - has been applied on numerous occasions to machine learning problems - see references by Vandereycken on matrix completion and by Sepulchre on matrix factorization. 

The computational cost of the two approaches is not carefully discussed. For instance, how expensive is the SVD in (7)? One can always perform an efficient low-rank update to the SVD - therefore, a rank one update requires O(nd) operations. What is the computational cost of each iteration of the proposed approach?

[Final Decision · Program Chairs · 06 Feb 2017]
**ICLR committee final decision**

The paper is mostly clearly written. The observation made in the paper that word-embedding models based on optimizing skip-gram negative sampling objective function can be formulated as a low-rank matrix estimation problem, and solved using manifold optimization techniques, is sound. However, this observation by itself is not new and has come up in various other contexts such as matrix completion. As such the reviewers do not see sufficient novelty in the algorithmic aspects of the paper, and empirical evaluation on the specific problem of learning word embeddings does not show striking enough gains relative to standard SGD methods. The authors are encouraged to explore complimentary algorithmic angles and benefits that their approach provides for this specific class of applications.